# CAR T-Cell Therapy for Cancer: Latest Updates and Challenges, with a Focus on B-Lymphoid Malignancies and Selected Solid Tumours

**DOI:** 10.3390/cells12121586

**Published:** 2023-06-08

**Authors:** Hiu Kwan Carolyn Tang, Bo Wang, Hui Xian Tan, Muhammad Adeel Sarwar, Bahaaeldin Baraka, Tahir Shafiq, Ankit R. Rao

**Affiliations:** 1Department of Oncology, Nottingham University Hospitals, Nottingham NG5 1PB, UK; ht8000@my.bristol.ac.uk (H.K.C.T.);; 2University of Cambridge, Trinity Hall, Cambridge CB3 9DP, UK; bw478@cam.ac.uk

**Keywords:** CAR T cells, solid tumour, immunotherapy, chimeric antigen receptor T cells

## Abstract

Although exponential progress in treating advanced malignancy has been made in the modern era with immune checkpoint blockade, survival outcomes remain suboptimal. Cellular immunotherapy, such as chimeric antigen receptor T cells, has the potential to improve this. CAR T cells combine the antigen specificity of a monoclonal antibody with the cytotoxic ‘power’ of T-lymphocytes through expression of a transgene encoding the scFv domain, CD3 activation molecule, and co-stimulatory domains. Although, very rarely, fatal cytokine-release syndrome may occur, CAR T-cell therapy gives patients with refractory CD19-positive B-lymphoid malignancies an important further therapeutic option. However, low-level expression of epithelial tumour-associated-antigens on non-malignant cells makes the application of CAR T-cell technology to common solid cancers challenging, as does the potentially limited ability of CAR T cells to traffic outside the blood/lymphoid microenvironment into metastatic lesions. Despite this, in advanced neuroblastoma refractory to standard therapy, 60% long-term overall survival and an objective response in 63% was achieved with anti GD2-specific CAR T cells.

## 1. Introduction

Cancer immunotherapy has transformed the contemporary management of patients with many advanced malignant diseases, forming an important addition to molecularly targeted, antiangiogenic, and conventional DNA-damaging treatments. However, even combination immune checkpoint inhibition in an immunologically ‘hot’ cancer such as melanoma does not lead to long-term survival in more than 60% of patients, and the toxicity of non-antigen-specific immune stimulation is not inconsiderable. Although a cellular therapy product for cancer (Sipuleucel-T for advanced castration-resistant prostate cancer) first received regulatory approval in 2010, the product is no longer available and was never widely adopted. However, regulatory approvals of CAR T-cell therapeutics targeting the B-cell antigen CD19 occurred in 2017, and the last 5 years have seen an expansion of the range of diseases amenable to treatment with this cellular therapy. In this review, we discuss the limitations of current immunotherapies, why new therapeutic approaches are needed, the biology and mechanism of action of CAR T-cell therapy, the clinical data for anti-CD19 CAR T-cell therapy, toxicity of treatment, and the future promise of this treatment for the common epithelial cancers. This review is highly pertinent especially in view of the emerging multiple advanced immunotherapy approaches for cancer, beyond checkpoint inhibitors.

## 2. Successes and Limitations of Contemporary Cancer Immuno-Therapeutic Approaches

Historically, treatment of malignant disease comprised surgery, radiotherapy, and chemotherapy with limited survival benefit, particularly with advanced, metastatic disease. Despite a durable response to interleukin-2 in about 15% of patients with advanced melanoma [1,2] and renal cell carcinoma [3], as well as the use of adoptive T-cell therapy in melanoma [4], formal recognition of the critical and central importance of antitumour immunity in cancer biology, as a hallmark of cancer, only occurred as late as 2011 [5].

However, remarkable progress has been made and immunotherapy has entered ‘centre-stage’ as a broadly applicable therapy able to achieve durable remissions in a substantial proportion of patients. These ‘modern’ immunotherapies enhance and unleash antitumour T-lymphocyte (and NK cell) responses by inhibiting the immune checkpoints CTLA-4 [6] and PD-1 [7], which are negative regulators of T-cell activation, using antagonistic monoclonal antibodies such as ipilimumab and pembrolizumab.

Long-term overall survival can be achieved in 54% of advanced melanoma patients with combined anti-CTLA-4 and anti-PD-1 blockade [8] although durable response is less common in mucosal [9] and ocular melanoma [10]. In melanoma, anti PD-1 monotherapy is a valid treatment option associated with a 35% chance of long-term survival and objective response in 40% [11].

In advanced renal cell carcinoma, frontline ipilimumab/nivolumab was superior to sunitinib (IMDC poor/intermediate risk) with a 5 year overall survival of 43% versus 31% [12]. In this setting, when anti-PD1 immunotherapies are combined with antiangiogenic treatment, an objective response rate of 71% and median progression-free survival of 24 months can be achieved [13].

Even in immunologically ‘cold’ cancers such as non-small-cell lung cancer and cholangiocarcinoma, the role of immune checkpoint blockade is now firmly established whether given as monotherapy or alongside cytotoxic chemotherapy. In advanced NSCLC without driver mutation, carboplatin/pemetrexed chemotherapy with concurrent and maintenance pembrolizumab achieved a 3 year overall survival of 51% and progression-free survival of 37% [14]. In advanced NSCLC patients without prior systemic therapy with high tumour PD-L1 expression, frontline pembrolizumab without chemotherapy led to objective response in 43% with a median progression-free survival of 10.3 months and median overall survival of 26 months. Health-related quality of life was better with immunotherapy and long-term follow up from this study found a 5-year overall survival of 26.3% with pembrolizumab [15,16]. In advanced, chemotherapy-resistant urothelial carcinomas, second-line atezolizumab improved long-term overall survival compared with second-line chemotherapy, with a 2 year overall survival of 23% and improved tolerability [17].

However, many challenges remain, and there are important limitations of the currently available immune checkpoint inhibitors. Although anti-PD1 monotherapy is generally better tolerated than conventional chemotherapy [18], immunotherapy treatment is not without risk, especially in view of the non-antigen-specific nature of immune checkpoint inhibitors which simply ‘take the break off’ the immune system. In fact, double-CTLA-4-knockout mice develop severe and fatal autoimmunity early in life [19]. Anti-CTLA-4 therapy is associated with a high risk of autoimmune toxicity that can be unpredictable and severe such as colitis [20], and all forms of anti-CTLA and anti-PD1 therapy are associated with a small but real risk of irreversible toxicities such as neurologic toxicities and type-1 diabetes mellitus [21]. These toxicities are increasingly relevant in view of the extended life-expectancy of these patients with immunotherapy. Patients with systemic autoimmune conditions requiring prior or current immunosuppressive therapy were excluded from the pivotal cancer immunotherapy trials and are at high risk of severe autoimmune exacerbations with checkpoint inhibitors.

In view of the potential toxicities of immunotherapy and, often, the availability of alternative treatment modalities (such as chemotherapy or molecularly targeted therapy), the development of predictive biomarkers of which patients are likely to benefit from treatment would be extremely useful, and it would allow those destined not to benefit to avoid futile treatment and unnecessary toxicity. Although PD-L1 expression on tumour cells/associated immune cells may predict response to anti-PD1 therapy and is a validated biomarker in certain settings such as upper gastrointestinal cancer and non-small-cell lung cancer, it is often not predictive [22]. The tumour somatic mutational burden is broadly correlated with the likelihood of clinical benefit from immune checkpoint inhibitors, due to increased neo-epitope load for T-cell recognition, across a broad range of tumour types; however, the development of TMB as a valid, robust, easily applied predictive biomarker in the clinical setting has progressed slowly [23]. Other potential predictive biomarkers, include the nature and composition of the faecal microbiome and tumour immune-signature profiles [24], although these are not in widespread clinical use.

The key challenge will be to understand resistance mechanisms and to target these, whether by simultaneous targeting of multiple checkpoints such as LAG-3 and TIM-3, or by seeking to interfere with the function of key drivers of tumour immune evasion such as downregulation of MHC class I molecules and upregulation of regulatory T-cell and myeloid-derived suppressor cell function [25]. Current immunotherapies augment the latter phase of antitumour immune responses and may neglect the inability of tumour to present antigen efficiently; the combination of local therapies to enhance antigen-presentation [26] and create a more immune permissive microenvironment in combination with immune checkpoint inhibitors requires further evaluation [27], aiming to convert ‘immune-cold’ tumours into ‘immune-hot’ ones. These local ‘immunostimulatory’ treatments may include viral immunotherapy such as T-vec in melanoma [28].

## 3. Principle of Adoptive T-Cell Therapy for Cancer and Biology of Chimeric Antigen Receptor (CAR) T Cells

### 3.1. Adoptive T-Cell Therapy

Leveraging the immune system to specifically target and destroy tumour cells has been the central dogma of anticancer immunotherapy. Over the last 30 years, a major form of cancer immunotherapy—adoptive T-cell therapy (ACT)—has been used to treat patients with, predominantly, advanced malignant melanoma in an era when melanoma was largely refractory to available systemic therapies such as chemotherapy. It is a form of ‘passive’ vaccination. In essence, tumour-specific cytotoxic T cells are infused systemically into patients aiming for tumour regression, alongside prior lympho-depleting chemotherapy (or historically, total body irradiation) to make ‘space’ for the transferred lymphocytes and, often, systemic interleukin-2 to sustain expansion and survival of the transferred cells [29]. Use of low-dose cyclophosphamide as part of the conditioning regimen may also selectively deplete CD25^+^ FOXP3^+^ regulatory CD4 T cells, creating a more immune-permissive environment [30]. The key potential advantages of this treatment strategy are that if long-term persistence (and development of a memory phenotype) of the tumour-reactive T cells could be achieved, then cure of advanced cancer might be possible, and potentially activated T lymphocytes could reach privileged ‘niches’ where conventional anticancer therapeutics struggled to penetrate [31].

The therapeutic potential of this approach was demonstrated as early as 1988 when a single-arm study found an objective response rate of 60% in chemotherapy-refractory metastatic melanoma patients treated with autologous tumour-infiltrating lymphocytes (TILs) without prior exposure to high-dose IL2 and 40% in patients who had previously received IL-2 [4]. Some responses were durable, lasting over 1 year. These response rates are far higher than those achieved with chemotherapy for melanoma. A more recent study, performed in Israel, found that 50% of advanced, chemotherapy and IL-2 refractory melanoma patients achieved radiologic response to short-term cultured bulk tumour-infiltrating lymphocyte therapy with a tolerable toxicity profile [32].

Whilst notable successes of the ACT approach in melanoma are well-described, the clinical data relating to this treatment regimen has almost been exclusively generated at a single United States centre (the National Cancer Institute, Bethesda, MD, USA) and there is an extreme paucity of randomised, comparative studies of adoptive T-cell therapy which remains important because the outcomes achieved may simply relate to patient selection rather than treatment efficacy.

The key challenges for the conventional adoptive T-cell therapy approach are the requirement for surgical access to at least moderate volumes of fresh primary/metastatic tumour tissue for preparation of tumour-infiltrating-lymphocytes, the potential induction of regulatory T cells within the TIL product during prolonged in vitro culture with IL-2, the complexities of screening the cellular product for tumour antigen reactivity and specificity, and the requirement that patients are fit for chemotherapy and often high-dose IL-2. It is also estimated that the rate of successful large-scale TIL culture (approximately 5 × 10^9^ T-cells) is only approximately 35%. Some of these challenges can be partially overcome by strategies such as the use of 4-1-BB co-stimulation during in vitro culture, selection of CD8-positive T cells, and the use of PD-1 blockade in vitro [33]. Additionally, patients with rapidly progressive, symptomatic metastatic disease often deteriorate and become ineligible for T-cell therapy during the prolonged T-cell culture periods required in the laboratory.

### 3.2. Biology of Chimeric Antigen Receptor T Cells (Figure 1)

Antigen presentation, predominantly by dendritic cells, guides the differentiation of naïve T cells into effector and/or memory T cells, as well as the balance between Th1 and Th2 T-cell responses. MHC (major histocompatibility complex) proteins present antigens in the form of peptide–MHC complexes to the T-cell receptor. In general, MHC class I molecules present short (8–10 amino acids) peptides that are derived from proteasomal processing of endogenous proteins to CD8^+^ cytotoxic T cells, and class II molecules present slightly longer (12–16 amino acids) peptides from exogenous proteins via the endo-lysosomal pathway to CD4^+^ helper T cells [34], although cross-presentation of exogenous antigen via the class I pathway does occur [35]. Full T-cell activation and effector function also require simultaneous ‘co-stimulatory’ signalling such as the CD28–B7 interaction, as well as the T-cell receptor signal [36].

T-cell receptor downstream signalling is initiated through engagement of a TCR with a relevant peptide–MHC ligand. Lck is recruited to the TCR complex which phosphorylates ITAM signalling motifs. By binding to phosphorylated ITAM motifs, Zap70 is recruited to the plasma membrane and its active conformation is stabilised. Thereafter, Zap70 can propagate and amplify signals from TCR, with specific phosphorylation of linker for activation of T-cells (LAT). LAT comprises four main Zap70 sites for phosphorylation: Y132, Y171, Y191, and Y226. Phospho-Y132 recruits PLC 1 to enable Ras/MAPK pathway activation, and the other three sites are responsible for the recruitment of Grb2 and Gads adaptors that bind SOS and SLP-76, which leads to Ras, Rac, and Rho GTPase activation [37]. Active Ras stimulates a kinase cascade, activating Raf, then MEK, and finally MAPK to regulate transcriptional activators such as Fos, Jun, and Myc, resulting in T-cell activation and cytokine secretion.

Chimeric antigen receptors (CARs) are recombinant (i.e., synthetic) T-cell receptors that usually identify cell surface antigens present in the natural state on the surface of tumour cells. In contrast to the typical, endogenous TCR, which attaches to HLA–peptide complexes, CARs bind to molecules that do not need peptide processing or HLA expression for recognition. Thus, CARs can detect antigens on any HLA background, unlike TCRs, which are MHC-restricted. Additionally, CARs can target cancer cells that have decreased class I MHC expression or defective proteasomal antigen processing, both of which are mechanisms that help tumours evade TCR-mediated immunity [38].

CARs consist of three distinct elements: an extracellular antigen recognition domain usually derived from a single-chain variable fragment (scFv) that originates from a monoclonal antibody, a transmembrane domain, and an intracellular domain that activates T cells known as CD3ζ (zeta) [39]. Although the current FDA-approved CAR T-cell products all employ an scFv antibody domain for antigen recognition, this is not the only strategy; in fact, this approach is potentially limited by the development of human anti-mouse antibodies if murine scFv is used, along with neutralising anti-idiotype antibodies even with humanised antibody sequences. The use of receptor ligands as a mechanism for tumour recognition is a highly promising approach [40], and a CAR T-cell product expressing the IL-13 protein (with E13Y mutation) for IL3Rα2 overexpressing glioblastoma has already undergone phase 1 clinical trial evaluation [41].

The transmembrane domain, which is the closest part of the endodomain to the membrane, comprises a hydrophobic alpha helix that stretches across the membrane. The stability of the receptor is linked to this transmembrane segment. If the natural CD3-zeta transmembrane domain is present, it may lead to the integration of the synthetic TCR with the native TCR. Currently, the CD28 transmembrane domain is the most robust receptor. The endodomain is the functional end and the most frequent component is CD3ζ which includes three immunoreceptor tyrosine-based activation motifs. Evolution of CAR T-cell technology can be summarised as first-generation CARs consisting of the CD3ζ alone, the second generation including additional costimulatory signalling domains (CD28 or 4-1BB), the third generation combining two costimulatory domains, (e.g., CD28 and 4-1BB) [42], and the fourth generation additionally encoding a proinflammatory cytokine such as IL-12 or GM-CSF to enhance the immunogenicity of the tumour microenvironment and potentially recruit other innate immune cells [43].

One of the most exciting developments in CAR T-cell biology is the potential use of logic-gated cellular control by, for example, substituting the conventional CD3-zeta domain with proximal T-cell signalling molecules which may restrict T-cell activation to encounter cells expressing higher levels of target antigen, or using NOT-gating and inhibitory domains to prevent or attenuate T-cell activation when there is co-expression of a ‘normal’ antigen [44].

Due to the ability of genomically unstable cancer cells to downregulate tumour antigens when faced with the ‘selective pressure’ of CAR T cells targeting a single antigen, multitarget CAR T cells have entered clinical development. These strategies may be simple, such as sequential administration of CAR T cells with two different specificities, transduction of the same cell population with two viral vectors, or more complex approaches such as the use of transgenes encoding two svFc domains [45]. However, the use of bi-valent or bicistronic constructs may be associated with lower transduction efficiency and CAR expression.

Preparation of clinical-grade CAR T cells for therapy typically starts with leuco-pharesis to obtain a sufficiently large number of peripheral blood mononuclear cells (without G-CSF mobilisation) followed by cryopreservation of these cells. After being thawed at the manufacturing facility, the cells are selected and activated using anti-CD3 and CD28 paramagnetic beads followed by transduction with a self-inactivating lentiviral vector encoding the transgene of interest. The transgenic T cells are expanded until a sufficient number for treatment is obtained—typically in the region of 300 million cells. The cellular product is evaluated in terms of the level of CAR expression by flow cytometry and the ability of the cells to produce IFN-gamma in response to tumour cell lines expressing the target antigen expressing cells prior to treatment [46]. Other viral vectors, aside from lentiviruses, can also deliver the CAR transgene to T cells such as gamma-retroviruses [47]. Although viral transduction of T cells has been widely employed, disadvantages of this approach include a risk of insertional mutagenesis and potentially tumourigenesis, and responses to the viral DNA may attenuate expression of the CAR construct [48]. However, novel, nonviral, gene-editing technologies are emerging such as the CRISPR/Cas9 system [49]. In terms of optimisation of the cellular product, it remains unclear which is the most efficient T-cell subset to be the substrate for transduction with the CAR construct (i.e., CD4, CD8, alpha/beta, or gamma/delta T cells), and this is an area of active research [50]. Of note, the use of non-MHC-restricted gamma/delta CAR T cells may potentially allow ‘off-the-shelf’ cellular therapy using allogenic CAR T cells expressing the gamma/delta TCR [51]. The use of invariant NK-T cells and NK cells as a substrate for CAR expression is also being explored [52].

## 4. Contemporary Successes of CAR T-Cell Therapy Targeting CD19 in Haematological Malignancies

Chimeric antigen receptor T cells (CAR T cells) targeting the B-lymphocyte maturation antigen CD19 have transformed the therapeutic landscape for patients with haematological malignancies across a wide range of age groups and histological categories. The US Food and Drug Administration (FDA, Silver Spring, MD, USA) and the European Medicines Agency have approved the use of CAR T cells in young patients (up to 25 years) with acute lymphoblastic leukaemia (ALL), adults with diffuse large B-cell lymphoma, and patients with multiple myeloma, as evidenced by various clinical trials. Nonetheless, toxicities related to CAR T cells are well recognised, including cytokine-release syndrome (CRS), immune effector cell associated neurotoxicity syndrome (ICAN), infection risk, B-cell aplasia, and consequent hypo-gammaglobulinaemia. Lastly, the challenges surrounding the logistics of implementation in aggressive diseases (i.e., the risk of clinical deterioration during T-cell preparation in the laboratory) and the cost of CAR T-cell infusion will ultimately determine the direction of research and development of CAR T-cell therapy for haematological malignancies.

ALL is the most prevalent malignancy in the paediatric population; in this group, cure rates are as high are 85% with conventional chemotherapy [53]. Although allogeneic stem-cell transplantation has an important role for consolidation, outcomes in adults and older children are suboptimal, and CAR T-cell therapy is an important treatment option in the refractory or relapsed setting in this patient group. As the first approved CAR T-cell therapy for paediatric and young adults with refractory or relapsing B-cell ALL [54], Tisagenlecleucel (tisa-cel) was shown to achieve a substantial chance of long-term overall survival (i.e., cure) in the ELIANA trial reported in August 2017. Among the 68 subjects enrolled in the phase 2 single-cohort multicentre trial, 80% of patients achieved remission at 3 months, with a 79% survival rate at 1 year follow-up. The final analysis of overall survival and progression-free survival found a 63% 3 year overall survival and 50% 3 year progression-free survival, suggesting that just over half of patients could be cured in the second-line setting [55]. Additionally, brexucabtagene autoleucel (brexu-cel) was approved for adults following the publication of the ZUMA-3 trial involving 55 subjects in October 2021. In this phase 2 single-arm multicentre trial, 71% of patients achieved remission at 16 months [56]. It was noted in this study, however, that two out of 55 patients receiving T-cell therapy died from treatment-related adverse events (septic shock and cerebral herniation), and that treatment was not without risk, although successful and safe allogeneic transplantation was feasible after CAR T-cell therapy.

CAR T-cell therapy also has proven value in treating adult patients with both high-grade and low-grade B-cell lymphomas. A substantial minority of diffuse large B-cell lymphoma (DLBCL) patients treated with conventional chemo-immunotherapy (typically R-CHOP) fail to achieve disease remission, resulting in high mortality rate, considering the aggressive nature of high-grade lymphoma. For those with lymphoma relapse after high-dose chemotherapy and autologous stem-cell support, cure is extremely rare, and prognosis is particularly guarded [57]. However, favourable results were demonstrated from the three clinical trials looking at CAR T-cell use in patients with relapsing and refractory high-grade DLBCL. In the phase 2 multicentre ZUMA-1 trial in 2017, 101 patients treated with axicabtagene ciloleucel (axi-cel) infusion showed a 40% complete response rate at 15 months. Similar results were reproduced in the phase 2a single-centre JULIET trial in 2018 with 40% complete response rate at 14 months. Overall survival rates of 52% at 18 months and 65% at 12 months were illustrated in the ZUMA-1 trial and the JULIET trial, respectively [58,59]. Lisocabtagene maraleucel (liso-cel) was also approved in 2021 as results showed 53% complete response at 18 months in the multicentre TRANSCEND trial [60]. However, it should be noted that there are no published comparative or randomised studies of CAR T-cell therapy in this setting, and a real-world retrospective study suggested that, after adjustment for pre-treatment prognostic characteristics, although response rate was higher for CAR T-cell therapy, the superiority of CAR T-cell therapy compared with conventional treatment was less marked in terms of progression-free and overall survival. Durability of responses was, in fact, similar with conventional ‘salvage’ treatment and CAR T-cell therapy [61].

In addition to the abovementioned high-grade lymphomas, CAR T-cell therapy has demonstrated efficacy and a significant role in low grade lymphomas as evidenced by some trials. The phase 2 Zuma-5 trial in March 2021 illustrated the efficacy of axi-cel to treat patients with follicular lymphoma. A total of 124 patients with follicular lymphoma showed an extremely high 94% overall response rate, 80% complete response and 93% overall survival rate at 19 months [62]. According to ELARA trial preliminary data, patients receiving tisa-cel for follicular lymphoma also showed a 65% complete response rate endorsing the clinical use of CAR T cells [63]. Whether CAR T-cell therapy achieves remissions or potentially cures patients with advanced low-grade B-cell lymphoma remains to be determined.

Furthermore, CAR T-cell therapy has promising prospects in patients with refractory or relapsing multiple myeloma. Idecabtagene vicleucel (ide-cel) received FDA approval in March 2021 following the marked clinical response in KarMMA trial of patients refractory to or relapsing after three prior lines of systemic therapy including a proteasome inhibitor [64]. A median progression-free survival of 8.8 months with a 73% response rate, 33% complete response rate, and minimal-residual disease negative status in 26% were demonstrated in 128 patients enrolled in the trial. In February 2022, ciltacabtagene autoleucel (cilta-cel) gained approval for similar indications after 97 patients achieved a complete response rate of 82.5% at 2 years follow-up according to the CARTITUDE-1 trial [65].

Although CAR T-cell therapy has become available for advanced B-cell malignancies, the regulatory approvals are based on single arm phase 2 studies without a control/comparator group and large scale; comparative trials are awaited with interest for confirmation.

## 5. Clinical Toxicities of CAR T-Cell Therapy and Their Management

With the increasing adoption of CAR T-cell therapies in treatment of multiple haematological malignancies, one of the fundamental challenges faced is the toxicity profile of this therapy. Broadly speaking, these can be dichotomised as ‘on-target’ and ‘off-target’ adverse effects. Off-target effects are uncommon due to improvements in and optimisation of the cellular development process in-vitro. However, ‘on-target, off-tumour’ effects remain a concern.

Cytokine-release syndrome occurs when CAR T cells engage with cancer cells, which triggers an inflammatory cascade and cytokine release [66]. It typically occurs later with CAR T-cell therapy than bispecific T-cell-engaging therapy, usually occurring within the first 7 days of infusion [67]. The key proinflammatory cytokines driving this pathophysiologic process are thought to be tumour necrosis factor-alpha, interferon-gamma, and IL-1-alpha. It appears that the critical cytokine released from the CAR T cells as the cognate target is engaged is granulocyte-monocyte colony-stimulating factor (GM-CSF). GM-CSF release in the tumour microenvironment directs macrophages, monocytes, and dendritic cells to produce the proinflammatory mediators [68]. If there is widespread systemic release of such cytokines, there is a resultant increase in vascular permeability, which can potentially lead to multiple organ failure (mainly circulatory/cardiovascular, respiratory, and renal) following an initial presentation with fever, hypoxia, and hypotension. Disseminated intravascular coagulation (consumptive coagulopathy) may occur, and CRS may overlap with the haemo-lympho-phagocytic syndrome. The majority of patients with CRS manifest with low-grade constitutional symptoms such as fever, myalgia, arthralgia and fatigue and have grade 1–2 toxicity not requiring specific intervention. More severe CRS is treated with corticosteroids, tocilizumab (IL-6 receptor antagonist), and anakinra (IL-1 antagonist), and the immediate availability of full intensive care facilities for organ support is vital. The use of corticosteroids is controversial, however, due to a high risk of apoptosis of the transferred T cells and potentially compromised treatment efficacy. Dasatinib, a tyrosine kinase inhibitor typically used to treat chronic myeloid leukaemia, may have a role in severe CRS by applying a temporary brake on T-cell proliferation and activation, which is reversible upon drug cessation [69].

Immune-effector cell-associated neurotoxicity syndrome (ICANS) is a severe subset of cytokine-release syndrome. Acute neurologic dysfunction, such as tremor, delirium, expressive dysphasia, headache, confusion, and focal deficits in rare cases, is a manifestation of increased blood–brain barrier permeability and cytokine entry (predominantly IL-6, IFN-gamma, and TNF-alpha) to the central nervous system. Fatal cerebral oedema due to ICANS has been described and probably affects 2–3% of patients receiving CAR T-cell therapy. Due to the inability of tocilizumab to cross the blood–brain barrier, high-dose parenteral corticosteroids are the cornerstone of management of ICANS, and occasional patients have required ventriculostomy to relieve raised intracranial pressure and even intrathecal chemotherapy such as methotrexate [70]. More recent data, however, perhaps suggest that low-level CD19 expression on brain cells adjacent to the vessel basement membrane walls in perivascular areas explains ICANS as an off-target effect [71].

Allergic (and occasional anaphylactoid) reactions to the T-cell infusion have been reported acutely with one patient (receiving mesothlin-specific CAR T cells for pleural mesothelioma) experiencing anaphylaxis and cardiac arrest within minutes of treatment in association with high serum mast cell tryptase levels [72]. Tumour lysis syndrome is also well described with CAR T-cell therapy.

## 6. The Future Promise of CAR T-Cell Therapy for Advanced Epithelial Malignancies

One of the key factors that has permitted the successful clinical development of anti-CD19 CAR T-cell therapy is the fact that the CD19 antigen is exclusively expressed on cells of the B-lymphocyte lineage, and even complete ablation of the B-cell compartment can be managed with intravenous immunoglobulin replacement, which minimises the risk of infection with encapsulated bacteria. Therefore, on-target toxicity is minimal with this approach.

Transferring CAR T-cell therapeutic technology to common solid tumours is challenging, and one of the key challenges is that of antigen selection. It is well established that the vast majority of human tumour-associated antigens (such as carcinoembryonic antigen, Her2, and glypican-3) are not truly or absolutely tumour-specific and, whilst amplified on malignant cells, are often detectable at very low levels on normal cells in a wide variety of organ systems. An early cautionary tale in the development of CAR T-cell therapy for solid cancers came in 2010 when a patient with chemotherapy-refractory advanced colon cancer was treated with a third0generation CAR T-cell product targeting Her2. Although a large number of transgenic T-cells were transferred [10], fatal pulmonary oedema occurred within days of T-cell infusion due to the expression of very low levels of Her2 on non-malignant lung epithelial cells and massive accumulation of activated T cells in the lungs [73].

Another key challenge for adopting the use of CAR T-cell therapy for solid cancers is that of accessibility of the T-cell product to the site of metastatic disease. In B-cell malignancies, the transferred CAR T cells naturally come into extensive contact with the target tumour cells in the blood stream and lymphatic systemic (including the spleen and bone marrow). However, for solid malignancies with organ metastasis in the liver and bones (for example), the vascular endothelium and the stromal extracellular matrix may limit ingress of T cells to the target cells [74].

In terms of using transgenic T-cell therapy for advanced solid cancers, it should be noted that there is already a strong precedent for using bispecific T-cell engagers (‘BiTEs’) or redirected T-cell therapy with tebentefusp (IMCgp100) in patients with advanced, metastatic uveal melanoma in the first- and second-line settings [75,76], although tebentefusp is a conventional pharmacologic drug that ‘redirects’ CD3 positive T cells to an HLA-A2 restricted class I epitope from the melanoma differentiation antigen gp100 rather than being a cellular product that recognises intact cell-surface antigen.

A phase 1 study of the feasibility and safety of CAR T-cell therapy for paediatric and young adult patients with neuroblastoma and osteosarcoma found that treatment was feasible and could be safely delivered [77]. The cellular product in this early-phase study was a third-generation CAR T-cell therapy targeting the mesenchymal tumour antigen disialoganglioside GD2 and co-expressing two co-stimulatory signalling domains (OX40 and 4-1-BB); the lympho-depleting conditioning regimen was cyclophosphamide. Preclinical work demonstrated that, although GD2 is a tumour-associated antigen, it is expressed at low levels in the adult brain, particularly in the cerebellum and peripheral nerves; mouse models of CAR T-cell therapy indicated that off-tumour fatal toxicity (encephalomyelitis) was common if the affinity of the anti GD2 antibody was very high [78]. Addressing these valid biological concerns, the CAR construct also had the novel addition of a suicide gene—the intracellular portion of the caspase-9 protein fused to a drug-binding domain from the FK506-binding protein. Expression of this transgene means that the systemic administration of a small inert biomolecule AP1903 is able to cause dimerisation of the caspase-9 protein with subsequent activation of the proapoptotic pathway and resultant death of the transgenic CAR T cells [79]. Therefore, at the first suggestion of any significant toxicity, particularly neurologic toxicity, the drug (AP1903) could be administered to cause death of the CAR T cells and prevent fatal toxicity. This study found that 76% of patients achieved stable disease, although all eventually progressed, and persistence of the transferred T cells was limited.

A more recently published study of the clinical use of this CAR T-cell construct (anti-GD2 with caspase-9 suicide switch and two co-stimulatory domains) showed real promise in terms of clinical efficacy. Successful CAR T-cell manufacture was achieved in all patients. Preconditioning chemotherapy was fludarabine/cyclophosphamide. In 27 paediatric patients with relapsed or refractory neuroblastoma, an objective response rate of 63% was achieved, and nine patients achieved complete radiologic response. In patients treated at the full dose of 10^7^ CAR-positive T cells per kg body weight, 3 year relapse-free survival was 36% and overall survival was 60%. One patient experienced high-grade toxicity and was salvaged by pharmacological activation of the suicide switch. In terms of CAR T-cell persistence, the median duration of persistence was 3 months, although this extended to 30 months in some patients [80]. By contrast, whilst being aware of the risks of cross-trial comparisons, a Japanese study found that 3 year progression-free and overall survival was 15% and 16%, respectively, in relapsed/refractory neuroblastoma patients treated with chemotherapy and autologous/allogeneic stem-cell transplantation [81].

In an attempt to reduce toxicity and optimise efficacy, loco-regional application of CAR T cells has also been developed as a therapeutic approach. In a small early-phase study of anti-CEA-specific CAR T cells given concurrently with selective intra-hepatic radiotherapy (radio-embolisation) in patients with CEA-expressing liver metastases (predominately originating from colorectal and pancreatic cancer), there was evidence of a biological effect with a reduction in PD-L1 expression levels and IDO expression in the metastatic lesions after treatment and evidence of biochemical (CEA) response in all patients. No high-grade cytokine-release syndrome or neurotoxicity was observed. However, despite the simultaneous use of SIRT, median and mean overall survivals were disappointingly short at 8 and 11 months, respectively, although the study population received an average of two lines of prior conventional systemic therapy [82].

Local and intra-lesional delivery of CAR T cells has also shown promise in the setting of multifocal relapsed glioblastoma. A patient with progressive glioblastoma after surgery, radiotherapy, and temozolomide chemotherapy, as well as an FGFR inhibitor, achieved nearly an 8 month tumour remission with complete response after treatment with IL-12Rα2-specific second-generation CAR T-cell therapy delivered directly into the postoperative resection cavity and intraventricularly. The patient was able to discontinue corticosteroids and achieved an improvement in performance status [83].

The CD133 antigen, a penta-span transmembrane glycoprotein, is a potential target for CAR T-cell therapy, being overexpressed in pancreatic and colorectal cancer and hepatocellular carcinoma, amongst others. It is a potential marker of cancer stem cells. A phase 1–2 clinical trial, conducted in China, of a second-generation anti-CD133 specific CAR T-cell therapy (expressing the CD3-zeta chain and CD137) produced using lentiviral transduction found encouraging insights that this form of cellular therapy could lead to clinical benefits [84]. The study population comprised 14 patients with advanced hepatitis B virus related hepatocellular carcinoma, seven with pancreatic adenocarcinoma, and two with colorectal adenocarcinoma. Tumours all expressed CD133 as assessed by immunohistochemistry. Patients had failed two prior lines of systemic therapy, and most of the HCC patients had bulky disease and portal vein involvement. The non-HCC patients underwent lymphodepletion with nab-paclitaxel and cyclophosphamide. Specifically, 62% of the CAR T cells were CD8-positive. T-cell dosing was 0.5–2 × 10^6^ CAR-positive cells/kg. Three out of 23 patients achieved partial radiologic response, and 14 patients achieved stable disease (lasting 9 weeks to 16 months). Median progression-free survival was 5 months for all patients and 7 months for HCC. A reduction in tumour burden of less than 30% (stable disease by RECIST criteria) was achieved in nine of 23 patients, and the vast majority of patients (21/23) did not develop new metastatic lesions on study. In seven patients, the CAR T-cell transgene was detectable for more than 8 weeks. However, overall survival was not reported in this study. As expected, early acute haematologic toxicity with pancytopaenia was the most common toxicity due to CD133 expression on haematopoietic progenitor cells [85].

Hepatocellular carcinoma, historically insensitive to conventional chemotherapy, continues to be a major healthcare concern, although the aetiologic drivers are changing with a reduction in hepatitis C virus-related cases and an increase in cases related to non-alcoholic steato-hepatitis. At least two-thirds of patients present with disease outside of curative criteria globally. There is emerging evidence, mainly from Chinese studies in patients with hepatitis B virus-related HCC, that CAR T-cell immunotherapy may be an important therapeutic modality for these patients especially in view of the recently demonstrated overall survival superiority of combined anti PD-1 targeted therapy and antiangiogenic therapy versus sorafenib [86]. Even with atezolizumab/bevacizumab, median overall survival is still only 19 months [87]; therefore, further therapeutic options are urgently needed.

A small study of 13 patients with incurable HCC, treated with a glypican-3 specific CAR T cells (lentivirally transduced, expressing CD3-zeta and a CD28 intracellular domain), provided revealing insights into the potential of this approach [88]. Glypican-3 is an onco-foetal antigen that is almost universally expressed on HCC and, importantly, not on dysplastic or regenerative liver nodules. The patients were all hepatitis B virus-positive and treated with entecavir, and the majority had extrahepatic spread; all had received prior local/regional and systemic therapy. All had Child–Pugh A liver function, and most were noncirrhotic. Most patients received 2 × 10^9^ CAR^+^ T cells per infusion. No IL-2 was used, and all except one patient had lympho-depleting chemotherapy with cyclophosphamide ± fludarabine. The median persistence of the CAR T-cell transgene in peripheral blood was 19 days although in one patient this was as long as 140 days. CAR T-cell persistence did not appear to correlate with clinical benefit. There was one treatment-related death due to severe cytokine-release syndrome and multiorgan failure despite corticosteroids and tocilizumab, although this was a patient with very-high-volume metastatic HCC. Median overall survival was 40 weeks with 1 year and 3 year survival probabilities of 42% and 10.5%, respectively. In terms of radiologic response, three of 13 patients achieved disease control, two had partial responses, and one had prolonged stable disease. These three patients also had major reductions in serum AFP levels.

Some realism, however, regarding the promise of the CAR T-cell therapeutic strategy comes from the setting of advanced pancreatic adenocarcinoma where conventional cytotoxic chemotherapy remains the standard of care with modest (approximately 6–9 month) gains in overall survival, and where immune checkpoint inhibitors, alone or with chemotherapy, have made no impact. An early study from Carl June’s group in Pennsylvania used mRNA electroporated CAR T cells transiently expressing an anti-mesothelin scFv alongside the CD3-zeta chain and the 4-1-BB intracellular domain to treat six patients with metastatic, chemo-refractory pancreatic ductal adenocarcinoma [89]. All patients had received at least two prior lines of chemotherapy. Ten patients were initially enrolled, but two did not receive T-cell therapy due to rapidly deteriorating during screening; one missed apheresis required for T-cell preparation, and, in one patient, T-cell culture and electroporation failed. The rationale for mRNA electroporation rather than conventional viral transduction was to transiently express the CAR construct to avoid off-target toxicity since mesothelin is expressed on normal pleura, pericardium, and peritoneum. No lymphodepletion was performed. There was no case of neurologic toxicity or cytokine-release syndrome. Two patients achieved stable disease for 3.8 and 5.4 months, and one patient demonstrated complete metabolic response of the hepatic metastases albeit with mild progression of the primary lesion in the pancreas. In terms of overall survival, the patient achieving 5.4 months of disease stability had an overall survival of 16 months, which is remarkable in the context of chemotherapy-refractory pancreatic adenocarcinoma.

However, a larger study of conventional lentivirally transduced anti-mesothelin specific CAR T-cell therapy (CD3-zeta, 4-1-BB) in serous ovarian cancer, pancreatic adenocarcinoma, and malignant pleural mesothelioma showed limited clinical activity [90]. Dosing ranged from 1–3 × 10^8^ to 1–3 × 10^9^ cells, and lympho-depletion with cyclophosphamide was performed. The median progression-free survival was 2.1 months, and only three of eight patients achieved stable disease for 3 months or more. No RECIST-defined partial or complete responses occurred. However, a key finding was that there was no on-target toxicity of pericarditis, pleuritis, or peritonitis; moreover, in three of five patients, the CAR transgene was detected in metastatic lesions suggestive of successful trafficking, particularly remarkable in the context of the dense stroma of pancreatic cancer. Following on from these findings, a mesothelin-specific CAR T cell expressing CD3-zeta, CD28, and the caspase-9 safety gene was evaluated as loco-regional therapy delivered intra-pleurally in patients with aggressive pleural malignancy (breast cancer, lung cancer, and mesothelioma) with cyclophosphamide preconditioning; remarkably, patients receiving concurrent anti PD-1 therapy with pembrolizumab achieved a median overall survival of 24 months [91]. Clearly, it is important to understand whether local delivery of the CAR T cells offers advantages over intravenous, systemic delivery alone in advanced solid tumours.

The setting of metastatic, castration-resistant prostate cancer is also a potential rich mine for CAR T-cell therapy, especially in view of the demonstrated overall survival benefit of a vaccination approach (Sipuleucel-T [92]) in this difficult-to-treat cancer. Prostate differentiation antigens such as prostate-specific membrane antigen are exclusively expressed on tissue of prostatic origin, potentially minimising off-tumour toxicity. A very recent phase 1 study of a PSMA-specific armoured CAR T-cell therapy, engineered to express a dominant negative transforming growth factor-beta (TGF-β) to evade immunosuppression in the tumour microenvironment, demonstrated clinical activity in advanced castration-resistant prostate cancer [93]. Three of 13 patients achieved a biochemical response (>30% reduction in serum PSA), and there was evidence of trafficking of T cells into metastatic lesions. However, one patient developed fatal cytokine-release syndrome.

## 7. Conclusions

Although progress has been made in the clinical application of CAR T-cell therapy in B-cell malignancies, even in this setting, there is a severe paucity of randomised comparative clinical studies, and the impressive results derived so far may relate partly to patient selection factors, as well as biological effect. It will be challenging to perform randomised trials of CAR T-cell therapy because manufacture of the T-cell product is not guaranteed (approximately 80% success rate), and application of the ‘intent-to-treat’ analysis (which is important for unbiased data interpretation) becomes problematic.

Despite several small-scale studies suggesting that CAR T-cell therapy is feasible and safe in solid tumours, the evidence of clinical efficacy remains low, especially in the commoner epithelial malignancies such as colon, lung, and breast cancer with median progression-free survival as low as 4–5 months. However, progress is likely, especially as the use of CAR T-cell therapy and other concurrent therapeutic strategies to target immune-regulatory mechanisms such as regulatory T cells, myeloid-derived suppressor cells, PD-L1, and IDO is explored. The combination of anti PD-1 therapy with CAR T-cell therapy may hold particular promise. Treatment-related mortalities are always a concern, even in patients with a limited cancer-related prognosis, but it is hoped that use of strategies such as the caspase-9 suicide switch within the CAR transgene will minimise this in the future. It will remain important to determine the exact contributions of lympho-depleting chemotherapy and systemic IL-2 (including dosing) to the therapeutic effect, particularly because many patients are heavily pre-treated and may not tolerate high-dose IL-2. Of particular relevance to solid tumours, novel routes of administration of the T-cell product such as intra-pleurally and via the hepatic artery may reduce systemic toxicity and improve efficacy. The generally limited persistence of CAR T cells and their questionable ability to differentiate into cells with a memory phenotype [94] are also issues, particularly in the context of potentially curative treatment. In conclusion, whilst an important addition to the cancer immunotherapeutic armamentarium, CAR T-cell therapy may ultimately only have a modest role in the common solid cancers, particularly in view of other emerging therapies such as bispecific T-cell engagers (redirected T-cell therapy [75]), the use of transgenic T-cell therapy, and locally delivered immune-stimulatory therapies.

## Figures and Tables

**Figure 1 cells-12-01586-f001:**
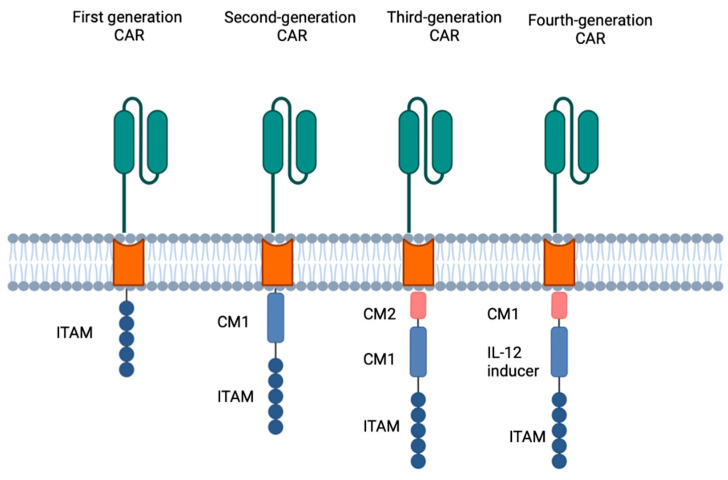
Four generations of CAR T cells.

## Data Availability

Not applicable.

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
