# Peer review of "CAR T-Cell Therapy for Cancer: Latest Updates and Challenges, with a Focus on B-Lymphoid Malignancies and Selected Solid Tumours"

_cells, 2023, doi:10.3390/cells12121586_

Round 1
Reviewer 1 Report
This is a very well written review that summarizes most of the key issues regarding the current status and problems associated with CAR T-cell immunotherapy. This reviewer believes it adds significant information for the medical community. There are, however, a few issues that need to be addressed including:
1. The Abstract and Introduction read almost identically and should provide separate information. I suggest the abstract be re-written.
2. In section 5 regarding clinical toxicities of CAR T-cell therapy and their management, the observations of the authors do not correlate with current literature. The authors state on lines 337 and 338 that the physiological process are TNFa and IFNg released by the activated CAR T-cells that activate nearby immune cells leading to host cytokine release. In fact, and more correctly as noted by Sterner et al., Blood 2019; 133(7):697-709 and Cox et al. in Leukemia, doi.org/10.1038/s41375-027-01572-7, the main culprit for the cytokine release is actually activate CAR T-cell secreted GM-CSF which then induces IL-1a, TNFa, and IL-6 host secretion by macrophages, antigen presenting cells, and monocytes.
3. Treatment of cytokine release syndrome is most often provided by anti-IL-6R antibody rather than as stated anti-IL-6. In fact, anti-IL-6 is often used as a second line of treatment if anti-IL-6R is not effective enough (see lines 349 and 350 on page 8.
4. Moreover, the neurotoxicity observed in some patients such as aphasia is more controversial since Parker et al. Cell 2020; 183:1-17 found that brain mural cells express CD19 which may be responsible for this adverse event often treated with steroidal therapy (see lines 36-365 on page 8.
5. The authors might also consider that lympho-depletion chemotherapy with cyclophosphomide and fludarabine may also deplete Treg cells to help the CAR T-cells retain their cytotoxicity when entering the tumor microenvironment. (see Becker and Schrama J. Investigtive Derm 2013:133(6); 1462-1468 and Zhao et al. 2010: 70(12):4850-4858.
Author Response
- The Abstract and Introduction read almost identically and should provide separate information. I suggest the abstract be re-written. Abstract has been completely re-written from scratch as recommended (150 words) without repetition.
2. In section 5 regarding clinical toxicities of CAR T-cell therapy and their management, the observations of the authors do not correlate with current literature. The authors state on lines 337 and 338 that the physiological process are TNFa and IFNg released by the activated CAR T-cells that activate nearby immune cells leading to host cytokine release. In fact, and more correctly as noted by Sterner et al., Blood 2019; 133(7):697-709 and Cox et al. in Leukemia, doi.org/10.1038/s41375-027-01572-7, the main culprit for the cytokine release is actually activate CAR T-cell secreted GM-CSF which then induces IL-1a, TNFa, and IL-6 host secretion by macrophages, antigen presenting cells, and monocytes. Agreed - these changes have been made, new reference added (Sterner et al)
3. Treatment of cytokine release syndrome is most often provided by anti-IL-6R antibody rather than as stated anti-IL-6. In fact, anti-IL-6 is often used as a second line of treatment if anti-IL-6R is not effective enough (see lines 349 and 350 on page 8. This has been changed as per the reviewer's note.
4. Moreover, the neurotoxicity observed in some patients such as aphasia is more controversial since Parker et al. Cell 2020; 183:1-17 found that brain mural cells express CD19 which may be responsible for this adverse event often treated with steroidal therapy (see lines 36-365 on page 8. This has been included with new reference.
5. The authors might also consider that lympho-depletion chemotherapy with cyclophosphomide and fludarabine may also deplete Treg cells to help the CAR T-cells retain their cytotoxicity when entering the tumor microenvironment. (see Becker and Schrama J. Investigtive Derm 2013:133(6); 1462-1468 and Zhao et al. 2010: 70(12):4850-4858. This has been updated to include a sentence on the potential selective depletion of regulatory T-cells by low-dose cyclophosphamide (ref Zhao et al).
Reviewer 2 Report
Here Tang et al provides a great introduction and update to the world of CAR-T therapies.
I strongly suggest the authors to reduce the length and readability of the second paragraph (Contemp canc immunotherap paragraph) as this does set the context of CAR-T therapy but is not the main focus of the review. By shortening this paragraph you can add a bit more information that is lacking:
Following the paragraph describing 1st-4th generation CARs, could the authors introduce the current trend of targeting multiple antigens either as bivalent CAR, 2 pools of CAR-T cells targeting multiple antigens, or multiplexed CAR with "OR", "AND", or "NOT" logic gating strategies.
Line 220-221 the authors make the statement that currently the CD28 tm domain is the most robust receptor. This is true when looking at their rapid and greater expansion, but the major issue is the lack of persistence due to exhaustion. This is even more apparent with CARs that have tonic signalling.
Could the authors elaborate a bit more on alternative cells for CAR therapy. A good overview of clinical trials with alternative cells is provided here: Beyond CAR T Cells: Other Cell-Based Immunotherapeutic Strategies Against Cancer - PMC (nih.gov)
Minor editing:
line 390 extra spacing between reference 60 and "lungs".
line 507: should be 4-1BB
Author Response
Here Tang et al provides a great introduction and update to the world of CAR-T therapies.
I strongly suggest the authors to reduce the length and readability of the second paragraph (Contemp canc immunotherap paragraph) as this does set the context of CAR-T therapy but is not the main focus of the review. By shortening this paragraph you can add a bit more information that is lacking: This paragraph, section 2 has been shortened and improved in terms of clarity as suggested
Following the paragraph describing 1st-4th generation CARs, could the authors introduce the current trend of targeting multiple antigens either as bivalent CAR, 2 pools of CAR-T cells targeting multiple antigens, or multiplexed CAR with "OR", "AND", or "NOT" logic gating strategies. Section on bivalent/bicistronic CARs has been added with appropriate references
Line 220-221 the authors make the statement that currently the CD28 tm domain is the most robust receptor. This is true when looking at their rapid and greater expansion, but the major issue is the lack of persistence due to exhaustion. This is even more apparent with CARs that have tonic signalling.
Could the authors elaborate a bit more on alternative cells for CAR therapy. A good overview of clinical trials with alternative cells is provided here: Beyond CAR T Cells: Other Cell-Based Immunotherapeutic Strategies Against Cancer - PMC (nih.gov). Section added on other cell types that can potentially used as part of CAR cellular therapy such as NK and iNKT cells with references.
Minor editing:
line 390 extra spacing between reference 60 and "lungs". - done
line 507: should be 4-1BB - done
Reviewer 3 Report
This review introduces the readers to the concept of immunotherapies, how they started and why they are needed. It aims to focus on solid tumors (as in the title) but extensively covers also hematologic malignancies. It mainly focuses on clinical results described in published papers.
Many reviews on CAR T cells are being published, owing to the rapid progress observed in the field. Some aspects covered by this review can be found in other reviews (introduction, CD19 CAR T cells), while other, specific to solid tumors and in particular epithelial malignancies are more original.
With some improvements and clarifications this review might be of interest to the readers of Cells. I personally miss some insight into novel development taking place at the pre-clinical level, but the decision to focus on the clinical aspect is also valid.
Points to be addressed:
1. Title should better reflect the content of the review. This review does not cover all solid tumors. It summarizes immunotherapies attempted for some solid tumors and provides an overview of clinical results in epithelial tumors (HCC, PC, CC) and also mentions the most recent results on GD2 in neuroblastoma.
2. References need to be formatted correctly
Line 44. Ref 1: I would cite the original work.
Line 45. References for renal cell carcinoma and ACT for melanoma are missing.
Lines 182-204 Description very detailed, not clear what the point is. If it is to introduce MHC-independent killing by CAR Ts then it is too much in my opinion.
Line 213 ff description of the extracellular recognition domain is not complete. Not only scFv are used. Receptor ligands can also be used. In the paragraph references are missing for the different statements. Endodomain description is also not complete. A suggestion: introduce also the more recent paper by Maizner at al describing novel endodomains https://doi.org/10.1038/s41586-023-05778-2 (given they were introduced already above...). This might revolutionize the CAR T field in the future.
Line 221 Not clear what “The scFv acts as the signal peptide of the ectodomain in a CAR.” should mean.
Line 230 Preparation of CAR T cells describes one historical example, but there are several different protocols in the clinic (i.e. retroviruses). Please complete or provide references.
Line 242 Reference for attenuation of CAR expression in response to viral DNA?
Line 243 CRISPR/Cas9 is strictly speaking not a gene transfer technology. It helps with integration in a specific safe loci, but not transfer into the cells. In the referenced paper thy also tak about “non-viral, gene-specific integrated CAR-T cells”
Line 332ff The authors introduces the concept of as ‘on-target’ and ‘off-target’ adverse effects. This is however misleading, because “off-target” implies effects resulting from activation of CAR T cells by engaging molecules other than the designed target (e.g. CD19). These effects can be largely controlled and eliminated during the development stage. What is more concerning, and can not be controlled are “on-target, off-tumor” effects.
In this context, how many of the observed adverse effects can be attributed to expression of CD19 in brain mural cells? https://doi.org/10.1016/j.cell.2020.08.022
Lines 393-394 The authors mention blood stream and lymphatic system as places where CAR T cells come in contact with malignant B cells. What about bone marrow?
Line 441 ff. Loco-regional application is a very important topic, and the author mentions intra-hepatic delivery. Other routes have been explored especially for brain tumors and might be mentioned here or do the author prefer to focus solely on epithelial malignancies?
Author Response
Points to be addressed:
- Title should better reflect the content of the review. This review does not cover all solid tumors. It summarizes immunotherapies attempted for some solid tumors and provides an overview of clinical results in epithelial tumors (HCC, PC, CC) and also mentions the most recent results on GD2 in neuroblastoma. Title has been changed - now refers to 'selected solid tumours'.
- References need to be formatted correctly - all references now correct
Line 44. Ref 1: I would cite the original work. Yes, reference added of original work with primary data in (Brown et al., 2008, CCR)
Line 45. References for renal cell carcinoma and ACT for melanoma are missing. These have been added and sorted
Lines 182-204 Description very detailed, not clear what the point is. If it is to introduce MHC-independent killing by CAR Ts then it is too much in my opinion. We feel this level of detail is reasonable here.
Line 213 ff description of the extracellular recognition domain is not complete. Not only scFv are used. Receptor ligands can also be used. In the paragraph references are missing for the different statements. Endodomain description is also not complete. A suggestion: introduce also the more recent paper by Maizner at al describing novel endodomainshttps://doi.org/10.1038/s41586-023-05778-2 (given they were introduced already above...). This might revolutionize the CAR T field in the future. We have added sections on logic-gated T-cells designed to reduced on-target, off-tumour toxicity based on the Maizner paper and also included a new section on the potential use of receptor-ligands rather than scFV domains for antigen recognition.
Line 221 Not clear what “The scFv acts as the signal peptide of the ectodomain in a CAR.” should mean. We agree this is ambiguous and this sentence has been removed.
Line 230 Preparation of CAR T cells describes one historical example, but there are several different protocols in the clinic (i.e. retroviruses). Please complete or provide references. Sentence about using gamma-retrovirally transuded CAR-T cells has been included with appropriate reference.
Line 242 Reference for attenuation of CAR expression in response to viral DNA? Reference has been added here - Atianand and Fitzgerald, 2013.
Line 243 CRISPR/Cas9 is strictly speaking not a gene transfer technology. It helps with integration in a specific safe loci, but not transfer into the cells. In the referenced paper thy also tak about “non-viral, gene-specific integrated CAR-T cells”. Agreed, we have changed gene-transfer to 'gene-editing'.
Line 332ff The authors introduces the concept of as ‘on-target’ and ‘off-target’ adverse effects. This is however misleading, because “off-target” implies effects resulting from activation of CAR T cells by engaging molecules other than the designed target (e.g. CD19). These effects can be largely controlled and eliminated during the development stage. What is more concerning, and can not be controlled are “on-target, off-tumor” effects. Additional sentences addressing this point have been added
In this context, how many of the observed adverse effects can be attributed to expression of CD19 in brain mural cells? https://doi.org/10.1016/j.cell.2020.08.022
This has been addressed - section added regarding low-level CD19 expression of brain mural cells as a potential explanation of CAR-T neurotoxicity.
Lines 393-394 The authors mention blood stream and lymphatic system as places where CAR T cells come in contact with malignant B cells. What about bone marrow? Agreed - bone marrow has been added (as part of the lymphatic systemic)
Line 441 ff. Loco-regional application is a very important topic, and the author mentions intra-hepatic delivery. Other routes have been explored especially for brain tumors and might be mentioned here or do the author prefer to focus solely on epithelial malignancies? Section added on use of CAR-T cells for advanced glioblastoma with treatment via intra-cavity and intra-ventricular routes.
Round 2
Reviewer 3 Report
The authors have addressed all the major issues, and this work is now suitable for publication. The readers of Cells will appreciate the detailed overview of current clinical trials and the insight into current status of CAR T cell therapy and potential for application in solid tumors.